# Adolescents Facing Transmedia Learning: Reflections on What They Can Do, What They Think and What They Feel

**DOI:** 10.3390/bs12040112

**Published:** 2022-04-16

**Authors:** Cinzia Runchina, Fernanda Fauth, Juan González-Martínez

**Affiliations:** 1Departament de Pedagogia, Universitat de Girona, 17004 Girona, Spain; cinzia.runchina@gmail.com (C.R.); fernanda.fauth@udg.edu (F.F.); 2Liceo Classico “G. M. Dettori”, 07029 Tempio Pausania, Italy; 3Research Group UdiGital.Edu, Departament de Pedagogia, Universitat de Girona, 17004 Girona, Spain

**Keywords:** transmedia learning, digital competence, secondary education, new media literacies

## Abstract

The integration of new media literacies and, consequently, strategies such as transmedia learning in the teaching–learning processes has been a topic of interest among various types of national and international institutions and governments. In this sense, the current article deals with the abilities, thoughts and expectations that Italian students in classical high schools have in order to face these new formative changes. For this purpose, a mixed methods approach (qualitative and quantitative) was designed and applied in the context of a classical high school in Cagliari (Italy): a questionnaire on digital skills (N = 128), a set of semi-structured interviews (N = 17) and two focus groups (N = 14). The results obtained allow us to verify that, from the point of view of skills, adolescents are prepared to take on the challenges of transmedia learning (navigation, information management), although their collaboration skills need to be strengthened. On the other hand, from the cognitive and affective points of view, they are positive and enthusiastic about these new possibilities: greater interaction, flexibility, engagement and variety of resources and learning strategies.

## 1. Introduction

In 2015 the Italian education system launched the Piano Nazionale Scuola Digitale (hereinafter PNSD), in step with many other initiatives throughout the Western world, with the aim of ensuring “the development of digital skills of students, with particular attention to computational thinking, the critical and conscientious use of social networks and media as well as production and links with the world of work”, as stated in Law 107/2015. With this, the PNSD aimed to concretize the conceptual frameworks of digital competences and literacies that the educational system must assume [1], and thus, the Italian school marked a forceful change from previous times and formulated a marked shift toward the integration of Information and Communication Technologies (ICT) in the educational practice [2] in a comprehensive and transversal way, so that it impacts on all learning processes and at all educational levels. The change is harmonious with what is happening throughout Europe and seeks to concretize the European Recommendation of 23 May 2018 on lifelong learning and the DigComp reference framework. Through them, it advocates educational policies that enable citizens to develop the digital competencies needed in the Knowledge Society [3].

This change, moreover, attempts to combat the traditional separation of the school and personal realities of adolescents, always straddling informal ways of learning that penetrate little into the school and vice versa [4] and that constantly prevent what is learned formally from becoming meaningful learning experiences and impacting on greater educational opportunities beyond school. In addition, we must consider the cultural and media context in which we live, characterized by participatory culture, media convergence and collective intelligence [5], in which we not only consume culture but also produce it collaboratively. A context, therefore, is needed where the informal and the formal should be communicating vessels [6] that allow us to effectively manage the challenges of learning.

With all this, transmedia learning (TL) offers suggestive opportunities, as it allows circumventing the boundaries between the formal and the formal. It allows adolescents a gradual step to participation (and citizenship) in a society that is digital and in which they should produce and not only consume, and do so in a critical, ethical and responsible way [7]. TL is a socioconstructivist and connectivist learning method that moves to the production of content structured by a story, in which the analog and the digital are mixed and alternated [8,9]. Moreover, transmedia (transmedia learning, in fact) is undoubtedly an opportunity from an inclusive and gender perspective [10,11].

However, both transmedia learning itself and participation in the new cultural and civic scenario we occupy (within formal or non-formal learning contexts) imply particular skills and abilities that go far beyond the concept of classical digital competence and lead us to ask ourselves whether Italian teenagers have the necessary new media literacies [12]. They allow media convergence and the leap from media consumption to creation (here, to learn), and one step further, we need to know what expectations and what implications such an educational change would have for the protagonists, adolescents.

## 2. Theoretical Framework

In this section dedicated to the theoretical framework, we will reflect on the concepts of transmedia, transmedia learning and transliteracy relevant to practical research. In the words of Pasteur, “Chance favors the prepared mind”.

### 2.1. Transmedia and Transmedia Learning

The first uses of the term *transmedia*, as we understand it here, were found in the reflections of Marsha Kinder from the 1990s [13], and they experienced a universal diffusion with the work of Henry Jenkins, within the fan culture and the emergence of media cultural phenomena in which consumers change to active participants in processes highly mediated by technology. All this flow gives rise to the concepts of media convergence and participatory culture [5]; the first one points to a context of overlapping and alternation of digital media, not linear or pre-established, but multiply branched and varied, and the second one to the possibility of users intervening in digital creation (thanks to the popularization of devices and the development of Web 2.0), which translates into processes of contribution, creation, and dissemination of content and, consequently, into users’ cultural practices (precisely with the converging media available to them). On the one hand, individuals go from being consumers to creators, and this creation is produced in a communal, not an individual, way (although with personal, not predefined, paths); on the other hand, cultural phenomena are conveyed in different media (so that the person who participates must be able to navigate between them, and in several of them, to be able to follow the flows of creation in which they participate and which they feed and not only consume).

From that reference, in the educational field there are different approaches to transmedia, in which a story or a narrative is the axis, and where it is developed across different (digital) media and with different participants. Thus, the term transmedia sometimes refers to a product [9,14], but it can also be a set of digital skills (knowledge and skills already acquired or to be developed) that the subject needs to participate in the creative process [7,15,16]. Finally, it can be a learning strategy (within a didactic approach related to storytelling in most cases), where students develop a story with different educational goals and where they must mobilize competences already acquired or develop new ones [8,15,17].

Regarding the definition of transmedia learning, no commonly accepted definition can yet be found. Transmedia learning rests on two main ideas: Jenkins’ concepts of participatory culture, collective intelligence and media convergence (the cultural framework) and the key roots of Vygotskian socioconstructivism [18,19] and Siemens’ connectivism [20] (the pedagogical framework). From that, we can design learning experiences guided by the need for the person to develop a narrative (fictional or not), with digital and non-digital resources of their choice and in a collective context or task (where the performance must be ongoing and learning is the result of collaborating with peers). Apart from these ideas, the theory highlights additional elements: connections with enactivism [21], ludic elements [22], and foreign language learning [17,23]. Because of this, transmedia learning is permeable and flexible, and very customizable: within the school and outside it, with different agents involved (families, teachers, educators), at every educational level (from childhood to higher education or non-formal education). In the end, we are talking broadly about processes of media convergence that encourage the active participation of users, who are considered prosumers (consumers and producers) of cultural/educational content, and these users are moved by their particular interests or passions across different media platforms to contribute to that narrative [17,24,25].

In particular, the opportunities of transmedia learning come precisely from its capacity for pedagogical principles that are not always easy to articulate (constructivism, connectivism) in motivating experiences (engagement) and that are customizable (the narrative allows layers, diversification, recurring points of connection) and coherent with the digital context we inhabit (but at the same time free of pressure from specific technologies and compatible with the analogical, from which we must not flee). Finally, this includes proposals that make it possible to offer new options to bridge, if desired, the limits of formal learning (in the broad sense) and of educational institutions (in particular).

### 2.2. Transmedia Literacy and Its Important Elements for Transmedia Learning

At this point, one of the key aspects we must consider is the nature of the learner (in any context) regarding the literacy they need to face and the learning opportunities they take advantage of. For the sake of economy of space, we will forego a broad reflection on the nature of the new citizens in the coordinates of the Knowledge Society, as interesting as it may be. In this sense, the literature already abounds in studies of the difference between the relationships that young subjects have with technology in their personal and academic spheres [26,27]. Some literature is also appearing on the differences that arise when learning is formal or informal [2,28]. However, it is necessary to define minimally what we mean by digital competence and what is different about this transliteracy. In this sense, if we start with the concept of media literacy we will understand in the Italian context the ability to use digital media and languages [6], to which is added, in a preferential way, the need to increase the skills of analysis, evaluation and critical reflection [29]. In the background is a myriad of reference frameworks and concepts of digital skills from which we will try to extract some transcendent and synthesizing notes [30,31,32].

Nevertheless, participating in the Knowledge Society requires different digital literacies (van Dijk 2017), and without them we face the risks posed by the different forms that the digital divide can take, which undoubtedly threaten adolescents’ empowerment as citizens (physical gaps in access, competence and use, according to van Dijk [33]), especially in a society characterized by media convergence and participatory culture [5], in which participation means not only consuming, but also actively producing. In fact, digital divides continue to exist, partly because of their diversity, heterogeneity and the multiplicity of forms they take [33]. Among them, we must pay special attention to the digital gender divide [34,35,36].

Regardless of the details of concepts and approaches, if this new media ecology (and TL) generates new ways of learning, we must also consider that there is a new and different literacy or set of skills necessary to successfully navigate across these media platforms and, at the same time, follow the thread of the development of the narrative. This can be called transmedia literacy (or transliteracy) [37,38,39,40], the components of which are yet to be developed. Within the new media literacies approach [12], some skills have been identified as necessary to live in the new digital cultural spectrum: transmedia navigation, game, performance, simulation, appropriation, multitasking, distributed cognition, collective intelligence, judgment, transmedia navigation, networking and negotiation. However, no prioritization is offered among them, nor is their special incidence from an educational perspective explored in depth [38]. These authors identify particularly important elements, such as transmedia navigation, prosumption (media consumption and production at the same time), collaboration and interaction (among peers), and, finally, the content curation.

### 2.3. Transmedia Learning

In recent years, some educational literature on educational technologies has focused on highlighting and exposing the gap between the literacy and informal learning practices of adolescents (e.g., video games, social networks, fan groups) and the formal learning processes within schools [4,12,41]. This gap undoubtedly points out the break between how adolescents learn (often via digital devices) in situations where they decide what to learn and why to learn it [4,42,43] and those where teachers design the instructional proposal in a completely institutionalized way [6], which does not necessarily succeed in translating adolescents’ learning efforts into meaningful learning [4,31,44].

However, schools should prepare them for life and, therefore, for being competent citizens, also digitally [3], in line with the European recommendations; therefore, we should focus on the digital (and transmedia) skills of Italian students, especially if we consider that TL can be an opportunity to effectively respond to the challenges ahead. Because of that, what we know of Italian adolescents’ digital profile is not enough, since it is too general: they are part of the most constantly connected population, and they show a complex and uniform pattern of digital consumption [45]. Of course, this is not always translated into a higher capacity or advantage when socially participating. Moreover, an entrenchment between the personal, family, and school worlds (and flows) can be always found: family and school worlds (regarding the digital issue) are vertical (from parents and teachers to adolescents) and unidirectional, and, in a different way in their personal and technological worlds, where interaction is always developed among peers [28].

Finally, from an inclusive education perspective, transmedia learning can be an opportunity from the point of view of universal design for learning (UD-L). The UD-L suggests that teachers, when planning learning activities, in general terms, respect three basic principles by offering multiple forms of representation, engagement, and action [46,47,48]. The alliance of TL and the UD-L philosophy can lead us to a more inclusive education [49,50,51] in terms of wider access to the curriculum [52], and digital and cultural accessibility [11,53,54]. This path will result in “the design and implementation of real educational proposals that respect the UD-L (*one size fits all*) principle and that, at the same time, allow not only the acquisition of digital skills necessary for the 21st century, but that can also contribute, at the same time, to dissolve or reduce the digital divide (including the gender gap) in inclusive education, remains an outstanding challenge. All of this, a priori, is at odds with the possibilities of transmedia learning” [32].

Transmedia learning, as we have seen, can be part of a wider change that disrupts this order, in line with the PNSD and the increasing social demands for changing the school. For this reason, it can be an opportunity to open the (psychological) barriers of the school and to involve in a more cohesive way all the elements of the adolescent’s ecosystem. Because of that, as a preliminary study anticipating a TL project implementation, the objectives of this study are, on the one hand, to identify the competences and transmedia profile of Italian students in the classical high schools, based on different instruments (digital literacy, transmedia profile, attitudes toward ICT), and, on the other hand, to determine the expectations and motivations of adolescents toward transmedia learning.

## 3. Materials and Methods

For this research, a mixed methods approach was considered, aimed at gathering quantitative (questionnaire) and qualitative (interviews and focus groups) information. This set of instruments (qualitative and quantitative) allowed us to extend (and in a complementary way) our knowledge of the object of study: how adolescents in the Italian classical high school face the challenge of transmedia learning, from the competence, emotional and rational points of view (what are their abilities, and what do they feel and think about it).

### 3.1. Participants and Context

Considering the universe of the classical high schools, it was decided to work with an accessible and incidental sample formed by students of all groups from any course in the classical high school G. M. Dettori in Cagliari (Italy), who agreed to voluntarily answer the questionnaire and participate in the interviews and the focus group. The fieldwork was carried out between December 2020 and November 2021. In relation to our sample, we consider this as one of the limitations of the research we present in terms of generalization. Since the sample in this analysis was not selected considering issues as equiprobability and representation (otherwise, our informants would have come from a real context), we cannot consider it is representative in its number or in its nature, taking adolescents’ general population as a reference. Therefore, the value of this research lies the novelty of the knowledge we can generate with these data and in the applicability of this knowledge (considering its usefulness as a diagnosis of the service of transmedia experiences in the concrete context of Italian classical high schools).

For those unfamiliar with the Italian educational context, the classical high school (*liceo classico* in Italian) is part of the Italian national system of public education based on the transmission of educational values aimed at forming citizens in the classical–humanistic tradition, which is its strong point. The cultural experience that the classical high school proposes is also aimed at enhancing logical–mathematical and scientific skills, skills in the field of foreign languages, art, and laboratory methodologies. The classical high school is not a vocational school (VET), so it is aimed at accessing universities and all the faculties available in universities. It has a two-year formative course and a three-year specialized course: in the two-year course the students take Italian, Latin and Greek grammar, Foreign Language, Geography, Mathematics, Science and, where there is experimentation, History of Art. In the three-year course, subjects such as Philosophy, Physics and the study of Classical and Foreign Literature are added. In the Italian education system, in addition to the classical high school, established by the Casati Law in 1859, there are other types of high school: scientific, human sciences, artistic, linguistic, musical and dance.

Regarding the questionnaire, we distributed the online version of the complete questionnaire (hosted in the university Google suite as a Google Form) to 138 students, 72 at face-to-face meetings and 66 in the online environment, by using the school learning environment platform (Italy was then in dual teaching, which means that every group was divided into two parts; one of them in person on some days at the school and the other following the classes by streaming, and vice versa; so those face-to-face and virtual meetings were held synchronously). As a result, we received 130 responses, representing different face-to-face/virtual sessions (using the scheduled group tutoring session for each of the school’s groups) and through an online form hosted on the university server, and 128 complete responses were consolidated (N = 128). As can be seen in Table 1, by gender, 68.7% were female (consistent with the larger female population usually found in Italian classical high schools), 21.7% were male and 1.6% chose not to be classified. By age, the sample was between 14 and 19 years old, with a regular distribution throughout the five courses students take during their experience in the classical high school, called *media superiore* in Italy.

Regarding the interviews, the population was represented by 17 students from different sections of the two-year and three-year periods of the same classical high school G. M. Dettori of Cagliari: men and women between 13 and 18 years old, all volunteers whose anonymity was guaranteed. Although a set of 20–25 interviews were initially planned, there were finally only 17 interviews because, as they progressed and a primary analysis was performed while transcribing them during the field work days, it reached saturation owing to the uniformity of many answers, and the research team decided to stop in an effort to interfere as little as possible with the rest of the school activities.

Finally, two focus groups were held to refine the conclusions drawn from the questionnaire and interview analyses. Fourteen students (6 in 2nd grade, 8 in 4th grade; 10 girls and 4 boys) participated in these groups, which were recorded and transcribed for analysis. The discussion was organized around a few thematic axes (displayed in the following section, with the rest of the information related to the instruments and techniques) in order to complete the general objective of the research, and it was moderated by a member of the research team, a high school teacher.

All students who participated in the three research techniques (questionnaire, interview, focus group) gave the corresponding informed consent, and at all times the authorization of those legally responsible for the school was obtained, as is legally required.

### 3.2. Instruments

For the quantitative part of this research, we first decided to use two existing instruments, both belonging to the conceptual sphere of media education and new media literacies. The New Media Scale was administered; this instrument develops Jenkins et al.’s (2009) categories, from which we chose 4 items (collective intelligence, judgment, transmedia navigation and visualization) [55]. We also applied the Media and Technology Usage and Attitude Scale [56], in order to characterize adolescents’ attitudes toward the use of technology in educational contexts. Finally, as to their level of digital literacy in a broad and non-specific sense, we also used the Digital Literacy Scale [57], created especially for European adolescents. All three scales are instruments that have been specifically validated and tested specifically for adolescent subjects in the reference studies. We summarize here some of their reliability characteristics and validation processes according to what has been already said for this set of instruments [32,58]. Regarding the Media and Technology Usage and Attitude Scale, its authors [56] conducted an initial literature review, a first pilot (N = 397) and the consequent re-vision and transformation of the items when needed, and a factor analysis that proved the final scale was internally reliable and externally valid. Regarding the Digital Literacy Scale [57], this author performed a pilot study (N = 208) to improve initial items and reduce the first version of the questionnaire, and a sequence of different steps (exploratory analysis, convergent validity scale test, confirmatory analysis), until she arrived at solid standardized regression weights for every subscale. Finally, the New Media Scale authors reviewed the literature, produced an initial version of the different items, piloted them (N = 397), performed a factor analysis in two steps, and reached a Kaiser–Meyer–Olkin test result of 0.824. This set of scales was administered in a unique way and by an online channel, at the request of the research team and under the auspices of the school’s management.

Regarding data treatment, a chi-squared test was applied to the non-parametric responses and an ANOVA test to the parametric responses; as for the establishment of the relationships between the different dimensions, we chosen the Pearson’s correlation coefficient. In both cases, confidence levels of 0.05 or 0.01 were used.

Regarding the qualitative part, interviews and a focus group were conducted to expand the general diagnosis offered by the quantitative data collected with the questionnaire and to deepen the dialogue with the adolescents about their conceptions and feelings about educational innovations with a high digital component such as the one we intended to implement. As to the interviews, participants were exposed to an interactive and dynamic talk about the challenges of education and the possibilities of transmedia learning, and from there they were interviewed. Semi-structured qualitative interviews were used, which included questions related to the objective of the research as well as to the selected variables of analysis (see Table 2).

The interviews were conducted in the school’s reading room at the disposal of the high school principal, with an average duration of 15/25 min. The interviews were audio-recorded with the prior permission of the students.

Finally, regarding the focus group procedure, the focus groups were conducted to complete the data that were collected following the administration of the questionnaire and the conducting of the interviews. They made it possible to identify the most relevant and recurring issues that emerged from the two meetings with the children. It should be noted that the children were sensitive to the subject in terms of digital literacy, and the enhancement of digital content, expressing a training need for themselves and for the school in general; children also want to confront each other and adults and to share information on a platform where information can be passed on. To achieve its goals, these topics and the research questions were presented to lead the sessions, as can be read in Table 3.

The recordings of these discussion interviews and focus groups were conveniently transcribed into textual format and imported into NVIVO 11, a program for qualitative data analysis. The data that emerged were transcribed, read, reread, coded for common relevant themes, and recurrent elements in the responses were coded in relation to the codebook provided for this purpose.

## 4. Results

For clarity of discourse, we will present the results sequentially, focusing on each of the techniques used: survey, interviews and focus groups.

### 4.1. Transmedia and Digital Profile

In terms of digital literacy, we found the values reflected in Table 4, in which we also incorporated the reference values [57]. As can be seen, in the following tables we decided to incorporate on the right-hand side the values of these same instruments in the research we took as a reference; thus, our data can be evaluated based on them (considering that the samples are not equivalent; despite the limitations of these comparisons, therefore, we considered that it may be useful to offer these values as well). While in some dimensions the values were slightly higher than those documented (for example, personal safety or critical skills), in others they were especially lower (above all, technological or information skills). In addition, the standard deviations were lower (in some cases considerably) than the reference values. From a transmedia perspective, this was positive in relation to the elements of transliteracy we saw that the literature highlighted as most important, since they emphasized the content elements over the technical elements.

Regarding the four dimensions of transmedia analyzed here (Table 5), the informants were especially inclined to transmedia navigation and to everything that had to do with the evaluation of the information found on the network, which can seem to be a contradiction of their documented values in the previous indicators. On the other hand, they presented more contained values both in the community dimension (collective intelligence) and in the assumption of other identities (visualization). Again from a transmedia perspective, this was positive indeed in relation to the highlighted elements of transliteracy (judgment and transmedia navigation). As can be seen, in this case we took as reference value empirical research [31] and not the text where the instrument was presented [55], since no general exploitation data were found in that text but the pilot.

Regarding attitudes, we found interesting elements when we compared our results with the reference values (Table 6): the dimension of positive attitudes was slightly higher than the reference values, and, in turn, the values of the dimensions of anxiety and dependence or negative attitudes were significantly lower. All of this made up an attitudinal profile favorable to ICT in a general way.

At the end of the quantitative part, we proposed an analysis of bivariate correlations between adolescents’ digital competence and the transmedia and attitudinal dimensions, which is shown in our last table (Table 7). All the transmedia dimensions analyzed were very significant and positive (more digital literacy also implies more propensity to transmedia), but, as we can see, no relation could be found between their digital literacy, on the one hand, and their attitudes toward technology, on the other, which is something to consider.

### 4.2. Regarding the Interviews

In this section, we will analyze the results of the interviews according to the three main thematic axes.

#### 4.2.1. Views on Technology-Mediated Learning

In general terms, we can say that adolescents are clear that learning processes should be mediated by technology: if technologies are omnipresent in life in general, they should also be omnipresent in education. They understand that technology brings value and, therefore, the school should be open to ICTs as resources and tools that can expand learning opportunities:


*Technology is a feature of the modern world and, therefore, must be used in the school in the right way, but one must know how to use it. [Int. 11]*



*For me the devices are very useful, even economically, cheaper than the textbooks we use in school or the vocabularies. [Int. 6]*


Hand in hand with these reflections always comes a certain censure for the lack of updating the school in general, and also specifically in relation to technological issues:


*In school we use technology too little and we should use it more because we are modern and we have to be up to date. [Int. 8]*



*The school, I think, should be more modern, more current, because today it is not like that. [Int. 3]*



*I think schools should have more technological resources, more computers, more computer labs. That is, here we have the interactive whiteboard in all the classrooms, but it is hardly used, but it is a computer, we use it, that is, we used it before when there was no COVID, at recess or in the time slots. [Int. 5]*



*Yes, school should be more modern, it should prepare us for tomorrow, […], instead it seems old to me. [Int. 2]*


However, there were also those who were carried away by tradition in their evaluation of the school and considered that a balance must be found between innovation and the maintenance of classical structures that do not allow the educational institution to lose its meaning. Implicitly, the two poles of two independent binomials (modernity–tradition, vanity–solidity) were set against each other and a sort of relationship was established between them:


*Yes, yes, of course it depends on the aspects because the school has to continue being an institution as before, but some things have to be improved, there are some retrograde aspects, but it has to continue being a serious place and some things can never change, in my opinion, because it would no longer be a school. [Int. 14]*


It is important to emphasize, in any case, that no negative considerations were made about the presence of technology in the learning processes in general, nor about its incursion into the school as they know it.

#### 4.2.2. Learning Practices

As for how they learned when they used technology, both inside and outside school, we found that in the informal context (unrelated to formal learning) it was common to integrate ICTs in the learning process, and it was difficult to comprehend that, when they were the ones who decided what they learned following their own interests, they did not place that learning in a technological context and did not use digital resources.


*Bah, I look for all kinds of information, about sports, especially, about the pandemic, about music, about technology, about school. [Int. 8]*



*For my own interest, different subjects, movies, music, small trivia. Now that we have started civics I need it a lot, for mathematics then it is clearer for me as I can watch and review videos. [Int. 13]*



*I am looking for a lot of information, about politics, about the municipality where I live, about Covid. [Int. 5]*



*Well, I do it for my personal interests, because if I’m curious to know something, I look it up immediately, it’s almost automatic. [Int. 16]*


They were aware that they learned in a different way than before (here there were constant references to how older people, for example, their parents, have learned and learn). They gave an important weight to the network as a virtual space in which learning takes place today, as opposed to spaces that they did not frequent (sphysical spaces), such as the library, and as opposed to analogical resources such as the encyclopedia or the book, which for them were not the first option (not even one of the most likely ones):


*Well, certainly, compared to my parents I will have learned things in a different way, especially with more influence from the internet and social media, because obviously, now, if I don’t know something I go to the internet, whereas before you would pick up an encyclopedia, open it and read. So learning is definitely different. [Int. 16]*


The assessment of these two ways of learning (the analogical way that they attributed to their adults of reference and the digital or hybrid one that they assigned as their own), evidently, was not neutral. They considered that the current way of learning (theirs) is much richer and more profitable:


*We learn in a better way because you have more tools anyway, more possibilities to take information, even before we used books, but now there is internet it is really complete and you take a quarter of the time than in the past. [Int. 15]*


Although we stated that adolescents focused on this “new way of learning” in the informal sphere, giving channel to their own interests, there were also communicating vessels for the learning that takes place in school (or related to school activities). In these cases, the relationship was ambivalent: on the one hand, it was difficult to resist the usefulness of the digital in formal learning, but it was also recognized that the digital was only part of what was necessary for studying:


*Yes, I think it is very useful, even when it comes to school, because, for example, when it turns out that I don’t know the meaning of a word, you just type it in Google and you find it immediately, but also in any subject you can find a lot of information. [Int. 26]*



*Eh, it is not essential, but the book is essential, then, if necessary, there is the internet, but only from the internet I could not study. [Int. 14]*


As for the problems they encountered in meeting their learning needs on the Web, it was common to identify the challenge of information evaluation: how to curate content, how to verify the quality of information, how to select relevant information (first excerpt). However, it seemed to be more linked to learning situations in school because, as seen in the second intervention, in the non-school environment these problems were much more easily solved:


*I practically never do it, that is, I take the information that interests me and that’s it. [Int. 3]*



*I have quite a few problems, because in several sites you find totally different information, so I compare myself with others, but it is difficult. [Int. 14]*


Finally, when asked to what extent in these technology-mediated learning processes they jumped from consumption to production, the answer was unanimous: they only consumed, and did not produce content (or at least they were not aware of it). They participated in the network, they commented and interacted, but they did not create.


*Consumption only. [Int. 8]*


#### 4.2.3. Educational Opportunities of Transmedia

Finally, as for the educational opportunities of transmedia, there was general and overwhelming enthusiasm. The TL briefings they received seemed suggestive and promising, and the perception was universal that it can open up an exciting range of possibilities for them. Among them, access to a seemingly infinite and immediate amount of information, presented in many different forms (complementary to each other):


*I think they would allow everyone to have immediate information, immediately visible, then there are the diagrams, the images, which are more fixed in the mind. [Int. 3]*


There were resources that complemented what happens at school, what is offered at school:


*Yes, I would say yes, I think the internet is unlimited, yes, it is useful to go deeper and it offers me, it suggests me ideas for what I study, I don’t limit myself only to what I hear in school. [Int. 3]*


On many occasions, however, these possibilities ended up materializing in ideas of a much lower profile than the vast range of transmedia (the use of digital resources that replaced analogical ones):


*I would say it would be great if they would let us use them, like the Latin and Greek dictionary, which is very heavy, mine is difficult to read because it was my mother’s, but that’s just a small example, there would be many. [Int. 11]*


In tune with the community essence of transmedia, they focused very strongly on the collaborative dimension of the opportunities offered in terms of interaction.


*Well, I could still ensure more collaboration between colleagues, maybe we all look for information together and then each one gives the information he/she found, so there could be more efficiency. [Int. 16]*


Only on rare occasions, and almost as a precautionary measure, were negative elements or reticence mentioned:


*I think so, but only in part, not all problems can be solved with technology, but in general I would say they are useful for some things. [Int. 7]*


### 4.3. Regarding the Focus Groups

In this section, we will analyze the results of the focus groups according to the two main thematic axes.

#### 4.3.1. General Evaluation of Transmedia Learning Opportunities

Finally, we come to the final part of the results, which has to do with the discussion carried out in the focus groups. From the conversations in relation to the appreciation of transmedia initiatives and their participation in them, a total homogeneity in the positive sense emerged, as stated by A (after each excerpt of the focus groups we indicate the coding of each informant with capital letters):


*Yes, I would say yes, because it was important to participate in order to understand how things are and also to be able to introduce technology in the school. (FG. A)*


In general, the possibilities of TL opened them up to a different way of learning, almost unknown until then.


*I was immediately interested in the initiative, also because it presents us with a way of learning in which I did not imagine that we students could participate. (FG. F)*


The students were explicit in pointing out the most innovative and interesting aspects that emerged from the interviews and questionnaire conducted in the previous phases of the project. The expectations, undoubtedly, could not be more positive, insofar as they allowed them to link what they do in school with their way of living and satisfied their knowledge needs outside school:


*Everything was new, it’s hard to say what was the most interesting.... well, maybe, I would say that the fact that we were presented with a very different way of experiencing school than usual, I would say that I never thought that the Internet in general had so much importance for our preparation. (FG. H)*


This linkage, in general, was very tempting to them because of its potential:


*But, basically everything got me involved, especially when I think about the fact that I wrote things about what I do with media outside of school, what I use, what I can be practically what I learn, that is, that I can learn also outside of school, with the different media out there, for so many topics, I mean, I had never thought about that. (FG. B)*


They also found the community dimension of transmedia learning suggestive, as was also apparent in the interviews:


*The interesting thing for me was to discover and understand that technology can be used at school to share with others, so also to socialize, to meet more people that maybe we can help and be helped. (FG. D)*


#### 4.3.2. Students’ Expected Involvement in a Transmedia Learning Project

Finally, we tried to expand through the information collected in the focus groups on the more subjective elements of their possible involvement in transmedia learning (motivation, expectations). In this sense, the students as a whole expressed different expectations of the implementation of the project and in terms of its usefulness for students with learning difficulties (in the key of inclusive education):


*It seems to me that this project can help some students who have difficulties in some subjects, because if you use different tools than the usual remedial ones in the afternoon, which are then only done if there is money, you can get results to help these students, with new things like that kind of learning. (FG. H)*


Students believed that there were new ways of learning that could be better adapted to their interests, to their ways of learning, and to the new resources that were now available to them and that were not as present in traditional schools as they would like. In this sense, they were clear that they would like to see them in practice, because they were suggestive:


*This project made me know unknown things, in fact I liked it and I would like to make it known in schools. (FG. E)*



*For me it was useful to understand that there are other ways of learning, which I practically did not know and which I would like to be present in the school. (FG. B)*


More importantly, they felt involved in this new way of approaching learning processes at school: to be taken into account in a much more digital, open and flexible school. They wanted what they used on a daily basis to penetrate the learning processes so that they could benefit from the opportunities it offered them.


*I like the idea of being part of the fact of learning, that technology helps in this sense, I would like this to be accepted in school, not hindered. (FG. B)*


They also found the community dimension of transmedia learning suggestive, as was also apparent in the interviews:


*The interesting thing for me was to discover and understand that technology can be used at school to share with others, so also to socialize, to meet more people that maybe we can help and be helped. (FG. D)*


## 5. Conclusions

The question we, as teachers, asked ourselves at the beginning of this reflection was whether the classical high schools could hopefully welcome the implementation of transmedia learning experiences, from the students’ perspective. This had to do both with their competencies and opinions (in relation to what the TL demands of them), and with the very nature of the classical high schools within the traditional Italian school system (indeed, they are classics not only because of the disciplinary field on which they focus, but also because of the weight of the Italian scholastic tradition, especially regarding the methodological issues). According to what we found in our mixed approach (not only with quantitative data from a survey, but also with qualitative data from interviews and focus groups), it seems that the conditions are favorable, insofar as students have the key skills to develop educational projects in transmedia and, moreover, as they seem motivated to do so.

In relation to the first issue, what we found is that the level of digital competencies in general is positive (dimensions such as critical skills or personal safety stood out, and informational or technological skills should be reinforced, although they allow us to be quite positive as well regarding the basis level), and, as for specific skills related to educational transmedia [39], we found particularly high values in transmedia navigation itself and in judgment (it was interesting to see how, although their levels of information literacy were not particularly high, those of this dimension were more oriented to the awareness of the need to evaluate information than to the evaluation technique itself); this is a very promising starting point for implementing transmedia learning activities with these students. However, of course, the landscape was quite heterogeneous regarding students’ digital profiles (in a general way) [31], and it will be necessary to accompany each of the students in the parallel development of the weaker dimensions (security skills, for instance, for every one of them, and in the development of weaker skills in every student). In any case, it will be necessary to always reinforce those aspects that are generally less strong in the context of the Italian school (the community dimension, collective intelligence; and visualization, linked to less traditional didactic strategies and less present in the school [42,44]).

As for how they felt about a transmedia learning proposal (and this is perhaps the most novel aspect of this research, although a priori obvious), adolescents were enthusiastic, and we saw this both in the quantitative data (high positive and low negative attitudes) and in the qualitative data (interviews and focus groups). Let us recapitulate the most important elements of what they think: their relationship with technology, despite all the reflections that have been made about it [27,31], is natural. They use technology in all dimensions of their lives and, therefore, also in learning situations outside school (when they learn by their own decision and by following their own interests); therefore, they understand that it is natural to learn also with technology. On the other hand, they feel that the school is, in methodological terms, an outdated institution that needs to be updated. They have a clear awareness of learning in a different way from their adults of reference—a much more efficient way of learning, thanks to the many resources now available to them (including technological ones). From here, it is logical for them to cook it all in one dish, in which the digital elements (and also the new learning strategies linked to the digital in the informal environment) enter the school to modernize it [42,59].

All this leads us to a final consideration. We started this paper with the concrete expectations of adolescents about a hypothetical implementation of TL experiences, and we see that the itinerary of research and knowledge construction (through the questionnaires, interviews and focus groups) generated motivating expectations for them: the school update [8], the variety of learning resources (in a UD-L approach) [11,46,48], greater interaction with their peers [9,25], engagement [14,23,38], etc.

After this general reflection on the data and their interpretation, we must conclude with two more important questions. The first has to do with the limitations of the study itself, directly related to the type of research carried out and the context. Although the results were clear and allowed for a solid interpretation, the qualitative part of the methodology applied in this context allowed for very cautious generalizations, and we must consider the context in the same regard (from a national or international perspective, considering the type of classical school, etc.), since this may have a very direct relationship with the findings of this research. On the other hand, these conclusions are aprioristic and allow us to think that transmedia learning experiences will be well received, but the practical implementation should also be monitored to determine what the students’ performance is in real cases and what their real satisfaction is a posteriori.

Secondly, in terms of implications, the results of this research precisely encouraged putting this step into practice, since the theoretical expectations about TL were in harmony with what students perceived and with the competencies they need: on the one hand we have skills and motivation, and on the other hand we have opportunities and potential. All this, undoubtedly, is the best scenario for jumping into practice (and further research).

It seems that here, therefore, we have the full circle, responding to what we were considering at the beginning of these conclusions (after responding, as we have seen, to the research objectives): the opportunities of TL a priori are innumerable and, moreover, not only do we have a concrete context in which to apply them (the classical high school), but its protagonists, the adolescents, would receive it with enthusiasm, convinced of the advantages that it can bring them in many ways.

## Figures and Tables

**Table 1 behavsci-12-00112-t001:** Characteristics of the participants (* official student age for each school year). Source: Authors’ creation.

Variable	Frequency	%
Gender		
Woman	88	68.7
Man	38	21.7
Non-binary gender	2	1.6
Course and age		
1st (14–15 years *)	22	17.2
2nd (15–16 years *)	27	21.1
3rd (16–17 years *)	37	28.9
4th (17–18 years *)	22	17.2
5th (18–19 years *)	20	15.6

**Table 2 behavsci-12-00112-t002:** Interviews. Research topics, questions, and objectives. Source: Authors’ creation.

Topic	Questions	Objectives
Views on technology-mediated learning	How does the learner deal with technology in learning contexts?	To explore the opinion of adolescents in relation to technology-mediated learning from a transmedia perspective.
Learning practices	How do they think they learn regarding technologies?	To know how they reflect on their own learning practices and what they think about that.
Educational opportunities of transmedia	What are transmedia learning practices?	To learn about technology-mediated learning practices linked to transmedia learning.

**Table 3 behavsci-12-00112-t003:** Focus groups. Research topics, questions, and objectives. Source: Authors’ creation.

Topics	Questions	Objectives
General evaluation of transmedia learning	What were the most interesting aspects of the project?	To know the students’ perspectives on transmedia learning.
Involvement	How do they feel challenged in relation to transmedia learning?	To know the implication that a transmedia approach to learning processes can generate in students.

**Table 4 behavsci-12-00112-t004:** Digital Literacy Scale. Source: Authors’ creation.

			Reference Values [57]	
	*Mean*	*SD*	*Mean*	*SD*
Technological skills	3.39	0.44	3.80	0.73
Personal Security skills	4.22	0.59	4.09	0.83
Critical skills	3.79	0.55	3.43	0.74
Devices Security skills	3.27	0.88	3.25	0.93
Information skills	2.56	0.67	3.37	0.70
Communication skills	3.61	0.55	3.69	0.58

**Table 5 behavsci-12-00112-t005:** Transmedia profile. Source: Authors’ creation.

			Reference Values [31]	
	*Mean*	*SD*	*Mean*	*SD*
Collective Intelligence	3.81	0.66	4.12	0.56
Judgment	3.96	0.50	3.88	0.59
Transmedia Navigation	3.97	0.67	3.75	0.70
Visualization	3.71	0.54	3.82	0.56

**Table 6 behavsci-12-00112-t006:** Attitudinal profile toward ICT. Source: Authors’ creation.

			Rosen et al. (2013)	
	*Mean*	*SD*	*Mean*	*SD*
Positive Attitude	3.71	0.51	3.66	0.84
Anxiety and depression	2.95	1.01	3.15	1.09
Negative Attitude	2.82	0.79	3.35	0.92

**Table 7 behavsci-12-00112-t007:** Correlations with digital literacy (** sig > 0.01). Source: Authors’ creation.

	C. of Pearson	Sig.
Positive Attitude	0.084	0.343
Anxiety and depression	0.059	0.507
Negative Attitude	−0.121	0.172
Collective Intelligence	0.251	0.004 **
Judgment	0.470	0.000 **
Transmedia Navigation	0.407	0.000 **
Visualization	0.463	0.000 **

## Data Availability

In this section, please provide details regarding where data supporting reported results can be found, including links to publicly archived datasets analyzed or generated during the study. Please refer to suggested Data Availability Statements in section “MDPI Research Data Policies” at https://www.mdpi.com/ethics. You might choose to exclude this statement if the study did not report any data.

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
