# Peer review of "Adolescents Facing Transmedia Learning: Reflections on What They Can Do, What They Think and What They Feel"

_behavsci, 2022, doi:10.3390/bs12040112_

Round 1

Reviewer 1 Report

These article about adolescents facing transmedia learning is attractive. The authors need to change or improve the next questions:

-Title: affirmative, because with the interrogation it seems less scientific, in this case (in other king of article it can be useful, but not in this case)

-Theoretical framework: all the tables need a source. The authors can include more updated citations.

-Methods. The combination of quantitative and qualitative tool is right.

-Results: it will be improved with more interpretation of the charts. Be careful with style (some mistakes introducing the Figures in the text)

-Conclusions: The authors can expand it.

Author Response

Dear reviewer,

thanks for your effort evaluating our paper. We appreciate every comment and suggestion very much indeed. Find attached the changes we have performed for every indication.

Comment

Change

These article about adolescents facing transmedia learning is attractive. The authors need to change or improve the next questions:

Thanks for your general comment.

-Title: affirmative, because with the interrogation it seems less scientific, in this case (in other king of article it can be useful, but not in this case)

We have changed it according to your advice.

-Theoretical framework: all the tables need a source. The authors can include more updated citations.

We have added the source to the tables and have included more updated references when possible.

-Methods. The combination of quantitative and qualitative tool is right.

Thanks.

-Results: it will be improved with more interpretation of the charts. Be careful with style (some mistakes introducing the Figures in the text)

We have reviewed and improved that section.

-Conclusions: The authors can expand it.

We have reviewed and improved that section as well.

Best regards,

the authors

Reviewer 2 Report

Regarding the content, I do not have any changes to recommend, it makes a good literary review to support the relevance of the problem to be studied and a good structuring of the content, it uses the correct methodology for this type of study and it is a consistent and well-detailed methodology to give significance to the results they show, makes a good discussion of the results with respect to the studies carried out previously, and marks the conclusion obtained well. Although I advise looking at these things: In section '5. Discussion and Conclusion' you should also add the implications of this study in the scientific world, the limitations of this study and future lines of work. Never two sections without a paragraph of text in between. You should put a couple of lines describing/naming the subsections you are going to deal with within that section. You must correct this between lines 60-61, 324-325, 356-358 and 492-493. Delete line 639. And the references in the 'References' section must follow the model set by the journal. You must correct the errors that exist. Look at this in the template.

Author Response

Dear reviewer,

thanks for your effort evaluating our paper. We appreciate every comment and suggestion very much indeed. Find attached the changes we have performed for every indication.

Comment

Change

Regarding the content, I do not have any changes to recommend, it makes a good literary review to support the relevance of the problem to be studied and a good structuring of the content, it uses the correct methodology for this type of study and it is a consistent and well-detailed methodology to give significance to the results they show, makes a good discussion of the results with respect to the studies carried out previously, and marks the conclusion obtained well.

Thanks for your general comment.

Although I advise looking at these things: In section '5. Discussion and Conclusion' you should also add the implications of this study in the scientific world, the limitations of this study and future lines of work.

We have added those elements to the final section.

Never two sections without a paragraph of text in between. You should put a couple of lines describing/naming the subsections you are going to deal with within that section. You must correct this between lines 60-61, 324-325, 356-358 and 492-493. Delete line 639.

We have solved those issues.

And the references in the 'References' section must follow the model set by the journal. You must correct the errors that exist. Look at this in the template.

We have reviewed them; we are also aware that a final revision of them will be also needed according to editor’s indications in the last version of the paper.

Best regards,

the authors

Reviewer 3 Report

It is an interesting study, but, in my opinion, it lacks the clarity that the reader needs. There are quite many things that need to be clarified / changed:

  1. It should be used, throughout the article, the term `classical high school` because the article is written in English. Consistency in terminology is needed (in the content of the manuscript you will find ‘licei classici’, ‘liceo classico’, `classical high school`. The terms in Italian language can eventually be used where you give explanations about the typology of high schools in the Italian national system of public education (section 3.1)
  2. Page 3:
  • row 94 (126) – what does it mean ‘presidential way’? Maybe ‘preferential’?!?
  • row 95 (127) – please change (Ranieri 2019) with its reference number and do the same throughout the manuscript.
  • row 107 (139) – what does it mean TL? Probably ‘transmedia learning’? Because somewhere in text above, you also talk about ‘transmedia literacy’. It is recommended to give the full name of a specific term and, eventually, put its acronym in parentheses.
  • row 133 (165) – it appears an ‘iwe’ should focus….
  1. Page 4:
  • row 149 (181)- please correct ‘can leads us to a with a…’
  • row 177 (209) – please correct ‘of all courses of any course’…
  1. Page 5 row 186 (218) - please correct ‘this these’ etc.

It is recommended that you read one again the text of the manuscript carefully and, if necessary, ask for an English speaker to help you.

  1. It is not specified in the manuscript how the questionnaire was distributed and to how many students? 130 answered from how many (what was the response rate, if it could be calculated)?
  2. In table 1 there in no 0.4% or 2 students (you started the phrase with ‘As can be seen in table 1 – page 6, row 236). Besides, in the first part of table (related to gender), the values ​​corresponding to the frequency must be reversed with those corresponding to the percentages (8 and 31.2 are the percentages, not the frequency). So please correlate the text with the table because otherwise the percentages are not correct.
  3. Page 6:

- rows 216-217 (248-249)- how many interviews were actually conducted from which you chose 17? Because you said ‘as they progressed, it has reached saturation due to the uniformity of many answers’. So, should we understand that there were several interviews and, due to the similarity of the answers, you took into account only 17 of them?

- Rows 219-220 (251-252): 10 girls + 8 boys = 18 not ‘Fourteen students’ (as you said in text, row 251)

  • Row 221 (253) you said: ‘was organized around the following thematic axes’? Which exactly? you need to reformulate
  • Row 232 (264)– please indicate the reference number for ‘Jenkins' et al. (2009)’
  1. Page 7:
  • Row 237 (269-270)– ‘All four scales’ – actually just above you talk about 3 scales ([54], [55], [56]).
  1. Table 4, 5 6 - the comparisons with the references are not very relevant because there were other subjects, other age groups, another nationality …. . Please indicate the references number. Why did you use the work Estebanell et al. (2021) – the reference [30] in table 5 as long as it is not among the 3 works you taken as a model for the used scales ([54], [55], [56])?
  2. In my opinion, a more concise, systematic presentation of the information in sections 4.2 and 4.3 should be offered to the reader. Page 15, the Section 4.3.2 appears twice (that section is effectively repeated).
  3. You talked about 2 focus groups (page 5, row 250). Therefore, what is the significance of the notations: FG.D, FG.H, FG.E ?? Do they stand for the focus groups D, H, E … or what those notations you used mean?

Author Response

Dear reviewer,

thanks for your effort evaluating our paper. We appreciate every comment and suggestion very much indeed. Find attached the changes we have performed for every indication.

Comment

Change

It should be used, throughout the article, the term `classical high school` because the article is written in English. Consistency in terminology is needed (in the content of the manuscript you will find ‘licei classici’, ‘liceo classico’, `classical high school`. The terms in Italian language can eventually be used where you give explanations about the typology of high schools in the Italian national system of public education (section 3.1)

We have harmonized it to classical high schools.

Page 3:

row 94 (126) – what does it mean ‘presidential way’? Maybe ‘preferential’?!?

row 95 (127) – please change (Ranieri 2019) with its reference number and do the same throughout the manuscript.

row 107 (139) – what does it mean TL? Probably ‘transmedia learning’? Because somewhere in text above, you also talk about ‘transmedia literacy’. It is recommended to give the full name of a specific term and, eventually, put its acronym in parentheses.

row 133 (165) – it appears an ‘iwe’ should focus….

We have added every suggestion according to your indications.

Page 4:

row 149 (181)- please correct ‘can leads us to a with a…’

row 177 (209) – please correct ‘of all courses of any course’…

We have added every suggestion according to your indications.

Page 5 row 186 (218) - please correct ‘this these’ etc.

It is recommended that you read one again the text of the manuscript carefully and, if necessary, ask for an English speaker to help you.

We have added every suggestion according to your indications.

We have reviewed all the text improving phrasing and spelling.

It is not specified in the manuscript how the questionnaire was distributed and to how many students? 130 answered from how many (what was the response rate, if it could be calculated)?

We have added that information.

In table 1 there in no 0.4% or 2 students (you started the phrase with ‘As can be seen in table 1 – page 6, row 236). Besides, in the first part of table (related to gender), the values ​​corresponding to the frequency must be reversed with those corresponding to the percentages (8 and 31.2 are the percentages, not the frequency). So please correlate the text with the table because otherwise the percentages are not correct.

We have reviewed and corrected those mistakes. Thanks for your careful reading.

Page 6:

- rows 216-217 (248-249)- how many interviews were actually conducted from which you chose 17? Because you said ‘as they progressed, it has reached saturation due to the uniformity of many answers’. So, should we understand that there were several interviews and, due to the similarity of the answers, you took into account only 17 of them?

- Rows 219-220 (251-252): 10 girls + 8 boys = 18 not ‘Fourteen students’ (as you said in text, row 251)

Row 221 (253) you said: ‘was organized around the following thematic axes’? Which exactly? you need to reformulate

Row 232 (264)– please indicate the reference number for ‘Jenkins' et al. (2009)’

We have reviewed and corrected those mistakes as well and added the missing notes you suggest. Thanks for your careful reading again.

Page 7:

Row 237 (269-270)– ‘All four scales’ – actually just above you talk about 3 scales ([54], [55], [56]).

We have reviewed and corrected that mistake.

Table 4, 5 6 - the comparisons with the references are not very relevant because there were other subjects, other age groups, another nationality …. . Please indicate the references number. Why did you use the work Estebanell et al. (2021) – the reference [30] in table 5 as long as it is not among the 3 works you taken as a model for the used scales ([54], [55], [56])?

We have explained that.

In my opinion, a more concise, systematic presentation of the information in sections 4.2 and 4.3 should be offered to the reader. Page 15, the Section 4.3.2 appears twice (that section is effectively repeated).

We have tried to rephrase these two sections, but in the end no substantial changes have been performed. We consider they offer a complete and detailed overview of these results which allow us to conclude as we needed. Our apologies.

You talked about 2 focus groups (page 5, row 250). Therefore, what is the significance of the notations: FG.D, FG.H, FG.E ?? Do they stand for the focus groups D, H, E … or what those notations you used mean?

We have explained that.

Best regards,

the authors

Reviewer 4 Report

Unfortunately, the general style of writing is characterized by the use of large sentences that make it difficult to read clearly. It is not clear in the Abstract what the authors mean by "mixed-methods approach" (line 8). Clause 2.2. titled "2.2. About transmedia literacy". However, the style of this title is difficult to combine with the scientific language of the text. The title of the paragraph should reflect the essence of the issue under consideration. Lines 200-206. It is required to rewrite this sentence, breaking it into several semantic sentences. The methodology must be clearly presented. lines 271-273. No need to save space. The validity of the characteristics gives the scientific validity of the article. The format of tables 3-5 is original. It might be better to use the classic table view.

Author Response

Dear reviewer,

thanks for your effort evaluating our paper. We appreciate every comment and suggestion very much indeed. Find attached the changes we have performed for every indication.

Comment

Change

Unfortunately, the general style of writing is characterized by the use of large sentences that make it difficult to read clearly.

We have reviewed all the text improving phrasing and spelling.

It is not clear in the Abstract what the authors mean by "mixed-methods approach" (line 8).

We have explained id better in the abstract.

Clause 2.2. titled "2.2. About transmedia literacy". However, the style of this title is difficult to combine with the scientific language of the text. The title of the paragraph should reflect the essence of the issue under consideration.

We have included your suggestion and have changed the title.

Lines 200-206. It is required to rewrite this sentence, breaking it into several semantic sentences.

We have reviewed all the text improving phrasing and spelling.

The methodology must be clearly presented. lines 271-273. No need to save space. The validity of the characteristics gives the scientific validity of the article.

We have improved the section.

The format of tables 3-5 is original. It might be better to use the classic table view.

We have explained it better.

Best regards,

the authors

Round 2

Reviewer 3 Report

Thank you for complying with the recommendations.

There are still some small corrections that should be operated to the manuscript’s text, but they will probably be made by the editors of the journal. For instance:

- at page 3, row 134, that „presidential” is still there, (van Dijk 2017) row 139 also …

- page 6 row 308, instead of „around the following thematic axes”, maybe it would sound better „around few thematic axes (displayed in the ….)”….

- the explanation on how the questionnaire was distributed is still quite elusive; preferably it would have been a form like: we distributed x questionnaires, y at face-to-face meetings, z in the online environment, by using the w platform. As a result we received 130 responses, representing …  I mean, in a scientific paper, it is recommended that everything to be very clear, very specific, without any ambiguities for the readers.

Author Response

Dear reviewer, 

thanks again for your careful revision. We have added the following changes:

  1. Preferential for presidential.
  2. Few for the following.
  3. We have added the information you suggested regarding the questionnaire distribution.

Best regards,

the authors

Reviewer 4 Report

In general, the authors corrected the comments.

Author Response

Dear reviewer, 

thanks again for your careful revision and for endorsing the manuscript.

Best regards,

the authors